# Meaning in Life Buffers the Association between Perceived Burdensomeness, Thwarted Belongingness, and Frequency of Non-Suicidal Self-Injuries in Spanish Adolescents

**DOI:** 10.3390/jcm10214867

**Published:** 2021-10-22

**Authors:** Jose H. Marco, Blanca Gallego-Hernández de Tejada, Verónica Guillén, Rosa M. Baños, Sandra Pérez

**Affiliations:** 1Departamento de Personalidad, Evaluación y Tratamiento Psicológico, Facultad de Psicología, Universitat de Valencia, 46010 Valencia, Spain; veronica.guillen@uv.es (V.G.); rosa.banos@uv.es (R.M.B.); Sandra.perez@uv.es (S.P.); 2CIBER Fisiopatología Obesidad y Nutrición (CB06/03), Instituto de Salud Carlos III, 28029 Madrid, Spain; 3Departamento de Personalidad, Evaluación e Intervención Terapéutica, Facultad de Psicología, Universidad Católica de Valencia, San Vicente Mártir, 46100 Burjassot, Spain; blanca.gallego@ucv.es

**Keywords:** meaning in life, perceived burdensomeness, thwarted belongingness, Spanish adolescents, non-suicidal self-injuries

## Abstract

Background: Adolescence is a developmental stage when there is a high risk of engaging in non-suicidal self-injury (NSSI). There is recent interest in the study of thwarted belongingness and perceived burdensome as variables associated with the frequency of NSSI in adolescents. Meaning in life (MIL) might be negatively associated with thwarted belongingness and perceived burdensomeness. To date, no studies have analyzed the buffering role of MIL in the association between thwarted belongingness and perceived burdensomeness and the frequency of lifetime NSSI in Spanish adolescents. Aims: (a) To test whether thwarted belongingness and perceived burdensomeness are associated with frequency of lifetime NSSI; (b) to test whether MIL moderates the association between thwarted belongingness and frequency of lifetime NSSI; and (c) to test whether MIL moderates the association between perceived burdensomeness and frequency of lifetime NSSI. Method: The sample consisted of *N* = 1531 participants (*n* = 736, 48.1%, were men, and *n* = 795, 51.9%, were women) between 12 and 18 years old from Spain. The participants filled out the Inventory of Statements about Self-Injury, Purpose-In-Life Test-Adolescent Version, and Interpersonal Needs Questionnaire. Moderation analyses were performed. Results: No statistically significant differences were found in the frequency of lifetime NSSI in the adolescents based on gender. Thwarted belongingness and perceived burdensomeness were positively associated with the frequency of lifetime NSSI in Spanish adolescents. MIL was a moderating variable between thwarted belongingness and the frequency of lifetime NSSI, and between perceived burdensomeness and the frequency of lifetime NSSI. Conclusions: Thwarted belongingness and perceived burdensomeness might be positively associated with the frequency of lifetime NSSI, and MIL might be negatively associated with the frequency of lifetime NSSI. Thus, adding these variables to current descriptive theories of NSSI in adolescents would allow us to improve assessment and treatment protocols for adolescents with NSSI.

## 1. Introduction

Adolescence is a developmental stage when there is a high risk of engaging in non-suicidal self-injury (NSSI) [1,2]. In fact, this behavior seems to be highly prevalent among college students, with prevalence rates ranging from 2.9% [3] to 42% [4], with an average of 18% across studies in different countries [5]. Longitudinal studies on NSSI in adolescence found that a history of NSSI is one of the most robust predictors of future NSSI [6,7] and suicide attempts [8,9] and found that the age at which an individual begins to self-harm is a potential risk factor for greater NSSI severity.

Several authors [10] suggest that there are four reinforcement processes through which NSSI are maintained. The four processes are configured in two dimensions: the interpersonal-intrapersonal dimension (referring to the nature of the consequent reinforcements) and the positive-negative dimension (referring to the type of reinforcement). Thus, by combining these two dimensions, four NSSI functions emerge: the positive reinforcement function-intrapersonal (e.g., self-stimulation), the negative reinforcement function-intrapersonal (e.g., alleviating a negative feeling), the positive-interpersonal reinforcement function (e.g., communicating to relatives that they are suffering and feel unhappy), and the negative-interpersonal reinforcement function (e.g., avoiding punishment by others).

In an effort to better understand NSSI, the Interpersonal-Psychological Theory of Suicide (IPTS) [11] could provide insight. Thwarted belongingness is defined as the perception of not belonging to a social group [12] and not having meaningful interpersonal connections [13]. Perceived burdensomeness is defined as the perception of being a burden to others [12] and being worth more dead than alive [13]. Drawing on the ideation to action framework, the IPTS suggests that three components must interact in order for a patient to develop a suicide attempt. First, passive suicidal ideation is caused by perceived burdensomeness and thwarted belongingness. This theory hypothesizes that if the patient feels that these two states will not change in the future, thus leading to hopelessness, active suicidal ideation will increase. Finally, to reach suicidal behavior, the patient must develop the capability for suicide through repeated experiences of pain, such as engaging in NSSI or exposure to violence [14]. In order to enact lethal self-harm behaviors, the patient must acquire the capacity to do so through repeated exposure to physical pain that produces habituation to pain. Participants who engage in NSSI report pain analgesia and tolerance during NSSI episodes [10,11]. Several studies have shown that an acquired capacity for suicide is obtained through the development of NSSI in adolescents [15].

However, although the IPTS does not suggest that thwarted belongingness and perceived burdensomeness are associated with NSSI, participants who engage in NSSI may have difficulties with emotional deregulation, interpersonal relationships, or impulsivity, which may lead to feelings of being a burden to their families [16]. Furthermore, people with NSSI indicated that they felt they were a burden to their relatives [17], leading to a feeling of perceived burdensomeness [12]. 

In the same way, individuals with a history of NSSI can lack meaningful social connections with others and feel that they are a burden to others. Previous studies found that people with NSSI had less interpersonal support than people without NSSI [18], and that low social support, isolation, and poor social interactions could make people with NSSI feel thwarted belongingness [12].

When people feel that they are a burden to other family members or that they do not belong, they experience deep emotional distress, which can lead to increased emotional deregulation and psychopathology. Performing NSSI would lead them to escape emotionally from this emotional distress, and NSSI would have the function of trying to bring emotional deregulation through emotional escape [19]. Thus, the NSSI would be maintained by negative intrapersonal reinforcement. Furthermore, the relationship between thwarted belongingnes and NSSI might also be maintained by the positive intrapersonal reinforcement function. For example, people who feel unaccepted and rejected and that they do not belong to anyone or anything can perform NSSI to receive more attention from the family, from psychiatrists in the emergency service, from psychologists in the hospital admission unit, and from the nurses who take care of their wounds. All this attention and care could increase their sense of belonging. In other words, after the NSSI, their sense of belonging could be increased. Thus, theoretically, based on the four-function model of NSSI [10], the association between the feeling of thwarted belongingness and perceived burdensomeness and NSSI could be maintained by means of negative intrapersonal reinforcement and positive interpersonal reinforcement.

To date, two previous studies have analyzed the association between thwarted belongingness and NSSI frequency. In a previous study with non-clinical adolescents found that lifetime NSSI frequency was moderately and positively associated with perceived burdensomeness and thwarted belongingness [17]. In the other study [13] in a sample of undergraduate students, found a low and positive association between lifetime NSSI frequency and perceived burdensomeness and thwarted belongingness. Therefore, there seems to be a certain consensus that thwarted belongingness and perceived burdensomeness are associated with lifetime NSSI frequency.

Nevertheless, the aforementioned studies focused on analyzing risk factors for NSSI, and no studies have analyzed protective factors between perceived burdensomeness and thwarted belongingness and the frequency of lifetime NSSI in adolescents. In this regard, the construct of thwarted belongingness could be negatively associated with the construct of meaning in life (MIL) [20]. MIL could be defined as the experience of freedom, responsibility, and self-determination, and he associates it with a positive view of life, the future, and oneself [21]. Other authors [22] suggested that MIL is composed of three dimensions: (a) coherence, the cognitive component of MIL, is defined as the degree to which people feel that the world around them is structured, predictable, and explainable; (b) purpose, the motivational dimension, refers to the way people experience that their lives are guided by valuable life goals; and (c) significance, the affective component, refers to the sense of the inherent value of life and implies having a life worth living. High levels of MIL contribute to proposing and achieving vital goals that orient and provide significance to one’s life. People with low MIL perceive their experience as fragmented and incoherent. They feel they lack life goals, nothing seems worthwhile in their future, and they perceive that their existence has little importance [23]. Thus, MIL is intrinsically associated with feeling connected to other people or to things (e.g., projects, goals, family, or friends) [21].

In adolescents, MIL has been positively associated with health, stress reduction [24,25,26], and positive affect [27], and it has been found to be a protective factor against substance use, risk behaviors [28], and hopelessness [29].

Regarding the association between MIL and NSSI frequency, several studies found that MIL was negatively associated with NSSI frequency in participants with borderline personality disorders [30,31] and eating disorders [32,33,34]. However, to date, no studies have analyzed the association between MIL and frequency of lifetime NSSI in adolescent participants.

In sum, taking the aforementioned studies into account, to date, no studies have analyzed the buffering role of MIL in the association between thwarted belongingness and perceived burdensomeness and the frequency of lifetime NSSI. Therefore, research is needed to better understand the possible association between perceived burdensomeness and thwarted belongingness and the frequency of NSSI [35], and it is necessary to analyze the role of hypothetical moderators (e.g., MIL) that might buffer the frequency of NSSI in adolescents [36]. Analyzing the possible associations between all these factors associated with the frequency of lifetime NSSI would allow us to improve current programs for the prevention or treatment of NSSI in adolescents.

Thus, the aims of this study are: (a) to test whether thwarted belongingness and perceived burdensomeness are associated with the frequency of lifetime NSSI; (b) to test whether meaning in life moderates the association between thwarted belongingness and the frequency of lifetime NSSI; and (c) to test whether meaning in life moderates the association between perceived burdensomeness and the frequency of lifetime NSSI.

Therefore, based on the studies mentioned above, we hypothesize that: (a) thwarted belongingness and perceived burdensomeness will be associated with the frequency of lifetime NSSI; (b) MIL will be a moderator between thwarted belongingness and the frequency of lifetime NSSI; and (c) MIL will be a moderator between perceived burdensomeness and the frequency of lifetime NSSI in Spanish adolescents.

## 2. Method

### 2.1. Participants

The study sample consisted of *N* = 1531 participants between 12 and 18 years old from Spain. The mean age was 14.85 years (*SD* = 1.56). Regarding gender, *n* = 736 (48.1%) were men, and *n* = 795 (51.9%) were women.

### 2.2. Procedure

The convenience sample was composed of adolescents between 12 and 18 years of age who were invited to participate in the research voluntarily through the schools. Participants and parents were given appropriate instructions to complete the assessment protocol. Parents were asked for written informed consent and a commitment to participate in the study, as well as the corresponding written authorizations for the minors’ participation, which were administered through the tutors of each course. An email address and a telephone number were provided to answer parents’ questions and help the participants. Only two participants contacted the researchers and were offered the resources they might need, and four parents were interested in knowing what questionnaires would be used and their possible risks. Participation was voluntary and anonymous, and no compensation was given for participating in this study. The inclusion criteria were that participants had to be adolescents between 12 and 18 years old, and the adolescents and their parents had to give their informed consent.

To recruit the convenience sample, 22 schools in Spain were contacted by mail or telephone. Eight of these schools finally decided to participate in the study. There are no differences between the schools that decided to participate and the ones that did not in terms of economic, cultural, and social characteristics. The participants were studying Compulsory Secondary Education or pre-university courses in Spanish schools: 25% of the participants came from the Valencian Community, 22% from the Basque Country, 17% from La Rioja, 20% from the Community of Madrid, 6% from Castilla la Mancha, 6% from Castilla-León, and 4% from Aragon.

The participants answered the questionnaires through an online platform on an ordinary school day in their own classes. The time period for answering the questionnaires was approximately 30–40 min. After the evaluation, a contact email was provided in case any of the participants required more information or needed to talk with a clinical psychologist. None of the participants made use of this service.

We followed the World Health Organization definition of adolescents as people from 10 to 18 years old. The study procedure was approved by the Ethical Committee of the Catholic University of Valencia Saint Vincent Martyr (Project Identification code UCV 2015-2016-25-V2).

### 2.3. Assessment

*Inventory of Statements about Self-Injury (ISAS)* [37]. The presence of NSSI was evaluated using this reliable and valid measure of NSSI frequency and functions. The first part of the inventory (*ISAS*-I) asks about the lifetime frequency of 12 different NSSI behaviors performed intentionally and without suicidal intent: banging/hitting self, biting, burning, carving, cutting, wound picking, needle-sticking, pinching, hair pulling, rubbing skin against rough surfaces, severe scratching, and swallowing chemicals. Adolescents were asked to indicate how many times they had injured themselves.

*Purpose-In-Life Test-Adolescents Version (PIL-A)* [29]. This scale is a 9-item version of the original PIL [38] for assessing meaning in life in adolescents. The items on the *PIL-A* are answered on a Likert scale ranging from 1 to 7, with a specific anchor for each item, and they assess two factors: satisfaction and meaning in life and purposes and goals in life. The total score ranges between 10 and 63. Higher scores represent higher meaning in life. The *PIL-A* assesses general beliefs about meaning in life, for example: “In life I have: from 1 = No goals or aims at all to 7 = Very clear goals and aims”; “I have discovered: from 1 = No mission or purpose in life to 7 = Clear-cut goals and a satisfying life purpose”. In the present study, the *PIL-A* showed high internal consistency (α = 0.89).

*Interpersonal Needs Questionnaire (INQ)* [39]. The *INQ* is a self-report measure composed of 15 items that evaluate the main constructs of interpersonal suicide theory (Joiner, 2005), which predicts the desire to commit suicide through two factors: thwarted belongingness and perceived burdensomeness. The items are answered on a Likert scale. Higher scores represent higher thwarted belongingness and higher perceived burdensomeness. Representative items from the *INQ* are: “These days, the people in my life would be better off if I were gone”; “These days, I feel disconnected from other people” (from 1 = Not at all true for me = 1 to 7 = Very true for me). The subscales showed good internal consistency for thwarted belongingness (α = 0.83) and perceived burdensomeness (α = 0.88) in our sample.

### 2.4. Statistical Procedure

First, means, standard deviations, and correlations between continuous variables were calculated. Second, we turned the *ISAS*-I into a 5-range scale: 1 = no presence of NSSI; 2 = between 1 and 4 times; 3 = between 5 and 50 times; 4 = between 51 and 100 times; 5 = more than 100 times. To test differences in the presence and frequency of lifetime NSSI between genders, we performed *t* tests and we used the Welch test when there was a violation of variance equality on Levene’s test. Next, we tested the moderation role of meaning in life (PIL) in the association between thwarted belongingness (PIL) and NSSI frequency. Finally, we tested the moderation role of meaning in life (PIL) in the association between perceived burdensomeness (*INQ*) and NSSI frequency. Moderation analyses were conducted using the PROCESS macro for SPSS32 [40]. To avoid multicollinearity, all the variables were mean centered prior to analysis. Potential multicollinearity between variables was rejected because the tolerance values were between 0.62 and 0.98 and variance inflation factors were between 1.61 and 1.05, which met good statistical criteria [41].

## 3. Results

Regarding the frequency of lifetime NSSI in the overall sample, *n* = 1115, 72.8% participants did not report any lifetime episodes of NSSI; *n* = 207, 13.5% between 5 and 50 episodes; 146, 9.5% more than 100 episodes; 32, 2.1% between 51 and 100 episodes; and 31, 2% between 1 and 4 episodes of lifetime NSSI. In the subsample of women participants, *n* = 587, 73.8% participants did not report any lifetime episodes of NSSI; *n* = 106, 13.3% between 5 and 50 episodes; 65, 8.2% more than 100 episodes; 22, 2.8% between 1 and 4 episodes; and 15, 1.9% between 51 and 100 episodes of lifetime NSSI. In the subsample of men participants, *n* = 528, 71.7% participants did not report any lifetime episodes of NSSI; *n* = 101, 13.7% between 5 and 50 episodes; 81, 11% more than 100 episodes; 17, 2.3% between 51 and 100 episodes; and 9, 1.2% between 1 and 4 episodes of lifetime NSSI. There were no differences in the frequency of lifetime NSSI by gender, *t*(1487,051) = 1.749, *p* = 0.08.

Table 1 shows the means, standard deviations, and zero-order correlations for the variables. The results indicate that thwarted belongingness was highly and positively associated with perceived burdensomeness (*r* = 0.62, *p* < 0.01), and it had a low association with NSSI frequency (*r* = 0.29, *p* < 0.01). Perceived burdensomeness had a positive and low association with NSSI frequency (*r* = 0.26, *p* < 0.01). Finally, MIL was highly and negatively correlated with thwarted belongingness (*r* = −0.61, *p* < 0.01) and perceived burdensomeness (*r* = −0.54, *p* < 0.01), and it had a low and negative association with NSSI frequency (*r* = −0.28, *p* < 0.01).

As Table 2 shows, thwarted belongingness (*t* = 4.53; *p* < 0.001; adjusted *R*^2^ = 0.09) and perceived burdensomeness (*t* = 4.63; *p* < 0.001; adjusted *R*^2^ = 0.07) predicted the frequency of NSSI in adolescents. Moreover, MIL moderated the association between thwarted belongingness and NSSI frequency (*F*_(5, 1525)_ = 39.90, *p* < 0.001, adjusted *R*^2^ = 0.12). After entering thwarted belongingness, MIL predicted NSSI frequency, both in addition to thwarted belongingness and when interacting with thwarted belongingness, thus supporting a moderating impact of MIL on the association between thwarted belongingness and NSSI frequency (Δ*R*^2^ = 0.01; *F*_(1,1525)_ = 9.39, *p* < 0.001). Figure 1 shows that, in adolescents with higher levels of MIL, increases in thwarted belongingness corresponded to smaller increases in NSSI frequency than in adolescents with low MIL.

Finally, MIL moderated the association between perceived burdensomeness and NSSI frequency (*F*_(5, 1525)_ = 36.28, *p* < 0.001, adjusted *R*^2^ = 0.10). After entering perceived burdensomeness, MIL predicted NSSI frequency, both in addition to perceived burdensomeness and when interacting with perceived burdensomeness, thus supporting a moderating impact of MIL on the association between perceived burdensomeness and NSSI frequency (Δ*R*^2^ = 0.01; *F*_(1, 1525)_ = 3.88, *p* < 0.001). Figure 2 shows that, in adolescents with higher levels of MIL, increases in perceived burdensomeness corresponded to smaller increases in NSSI frequency than in adolescents with low MIL.

## 4. Discussion

Regarding the first aim, our results suggest that thwarted belongingness and perceived burdensomeness were associated with the frequency of lifetime NSSI in Spanish adolescents, and that the variance explained by both was similar (9% for thwarted belongingness and 7% for perceived burdensomeness). Thus, our hypothesis was confirmed. Our results are similar to those found in other studies with adolescents, showing that the relationship between thwarted belongingness and perceived burdensomeness was associated with NSSI frequency [35], shedding light on the relationship between these variables in Spanish adolescents.

Regarding the second and third aims, our results suggest that MIL was a moderating variable between thwarted belongingness and the frequency of lifetime NSSI, as well as between perceived burdensomeness and the frequency of lifetime NSSI, confirming our hypotheses. However, although meaning in life was a moderator variable, the amount of variance explained by the models was small (*R*^2^ adjusted range 10–12%). Our results might be explained by the fact that adolescents with low MIL could develop a greater number of interpersonal and behavioral problems in the family, such as substance use, eating disorders, and risk behaviors [28,34]), which would increase their feelings of perceived burdensomeness, thwarted belongingness, and hopelessness [29]. These feelings, in turn, could lead to the use of maladaptive coping strategies with emotional deregulation, such as NSSI [42]. However, if adolescents with low MIL could discover and achieve long-term goals related to the main sources of MIL, such as good family relationships [43], good interpersonal relationships, health, personal growth [44], loving someone, or participating in activities with the possibility of building something unique and creative [21] 2006), their perceived burdensomeness, thwarted belongingness, as NSSI behaviors might decrease [32].

Moreover, our results are consistent with studies showing that connectedness was inversely associated with NSSI [45]. Klonsky and May [46] define connectedness as a personal connection to other people, other interests, roles, projects, or any sense of purpose or meaning that keeps one invested in living [47]. However, although connectedness is a similar construct to MIL, this construct only refers to two of the three dimensions of MIL [22]. Connectedness refers to purpose and significance, but it does not include the dimension of coherence. Recent studies with participants with borderline personality disorders suggest that it is necessary to consider all three dimensions of MIL in the association between MIL and borderline personality disorders, such as NSSI [48].

Although there is consensus about the moderating role of MIL in the risk factors for suicide in previous research with adults [49], we should highlight that this is the first study to support the moderation role of MIL between the main constructs of IPTS [11] and frequency of lifetime NSSI in adolescents. Our study did not include all the components of IPTS because we did not assess suicidal ideation or acquired capability for suicide. However, previous studies have shown that an acquired capacity for suicide could be obtained through the development of NSSI in adolescents [15]. Thus, future longitudinal research will be necessary to study whether MIL moderates the association between the components of IPTS and suicidal ideation or suicide attempts.

Regarding the frequency of lifetime NSSI, we found that 27.2% of the participants had carried out lifetime NSSI. Although there is high variability across studies regarding the frequency of NSSI [50], our percentages are higher than international review studies that found a prevalence of 16–18% [5]. However, our prevalence is lower than in the other study carried out in a Spanish adolescent sample (32%) [51]. This variability in the results could be due to several factors, such as the way NSSI is evaluated or the sociocultural characteristics of each sample.

Our results have important theoretical and practical implications. Based on our results, thwarted belongingness and perceived burdensomeness could be positively associated with the frequency of lifetime NSSI, and MIL could be negatively associated with the frequency of lifetime NSSI. Based on the theory of the functionality of NSSI [10], an adolescent who feels thwarted belongingness could perform NSSI in a way that mobilizes the environment and brings it closer to him or her, and this would be an interpersonal function that would maintain the NSSI. In the same way, from the IPTS perspective, an adolescent who perceives him/herself as burdensomeness could begin to develop passive suicidal ideation, thus increasing the possibility of developing NSSI. Performing NSSI would increase the feeling of being a burden to caregivers, which would create a vicious circle that would drive suicidal behaviors. We suggest that adding these variables to current descriptive theories of NSSI in adolescents would allow us to improve assessment protocols for adolescents with NSSI. It would be important to evaluate thwarted belongingness and perceived burdensomeness to detect possible risk factors for NSSI and suicidal behaviors because these two constructs are related to both. In addition, our results suggest the need to develop psychotherapeutic interventions for adolescents with NSSI that focus on orienting their lives towards goals that allow them to discover a meaningful life worth living. Meaning-focused therapy has been shown to be effective in studies with adolescents, increasing their meaning in life, reducing depression, and improving physical and emotional well-being [52]. Thus, a future line of research would be to carry out a randomized controlled study to verify whether therapy focused on meaning is effective in reducing the current frequency of NSSI, perceived burdensomeness, and thwarted belongingness.

Our study has some limitations that should be taken into account when interpreting the results. An important limitation is that the outcome measure was the frequency of lifetime NSSI, whereas the predictor variable *INQ* refers to participants’ recent perceived burdensomeness or thwarted belongingness. This is a key limitation given the lack of temporal ordering, and it is possible that the perceived burdensomeness or thwarted belongingness was a consequence of NSSI, and not the other way around. Previous studies suggested that people with NSSI had difficulties with emotional deregulation, interpersonal relationships, or impulsivity, which lead to feelings of being a burden to their caregivers [16]. In addition, people with NSSI showed a lack of meaningful social connections, isolation, and poor social relationships, which could make them feel thwarted belongingness [12]. This is a retrospective and cross-sectional study, which limits the conclusions that can be drawn about causality and directionality between the variables. For this reason, future research should confirm these results using a longitudinal design with a randomized sample and evaluate current self-harm. Second, recruitment for the study was nonrandom, and selection and self-selection bias may have affected the findings and, thus, the ability to extrapolate the findings to the population of interest (i.e., Spanish adolescents). Third, we did not assess data about former or present psychopathology, suicidality, depression, psychological treatments, mental illness, pharmacotherapy, drug intake, family situation, family support, or functioning in peer relationships, and so we did not control the role of these variables in the frequency of lifetime NSSI. Therefore, these variables could explain part of the variance we attributed to thwarted belongingness, perceived burdensomeness, or meaning in life. Moreover, although the sample was large, it was a convenience sample, and so the results found in our study cannot be generalized to the entire population of Spanish adolescents.

In conclusion, based on previous studies [13,17] and the present study, thwarted belongingness and perceived burdensomeness could be positively associated with the frequency of NSSI, and MIL could be negatively associated with the frequency of NSSI [30,31]. Thus, adding these variables to current descriptive theories of NSSI in adolescents would allow us to improve assessment and treatment protocols for adolescents with NSSI.

## Figures and Tables

**Figure 1 jcm-10-04867-f001:**
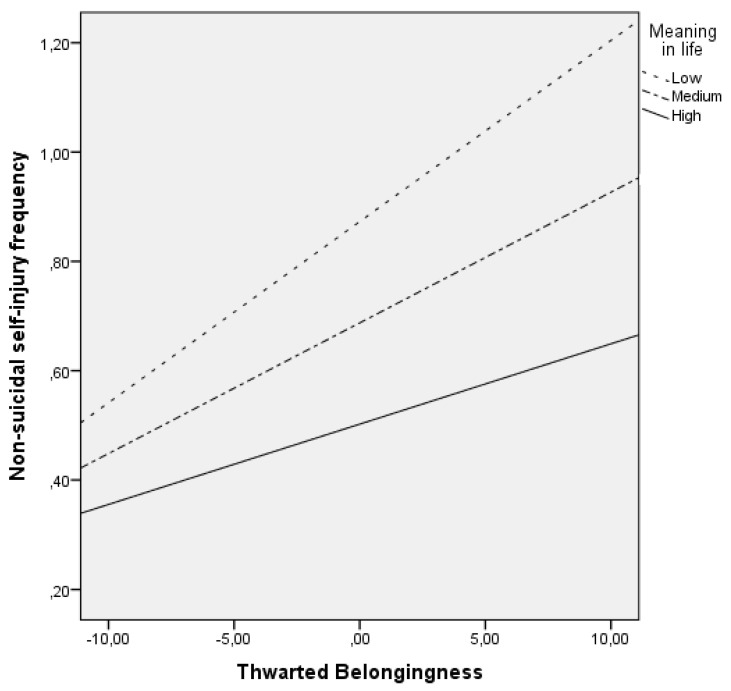
Meaning in life moderates the association between thwarted belongingness and frequency of non-suicidal self-injury.

**Figure 2 jcm-10-04867-f002:**
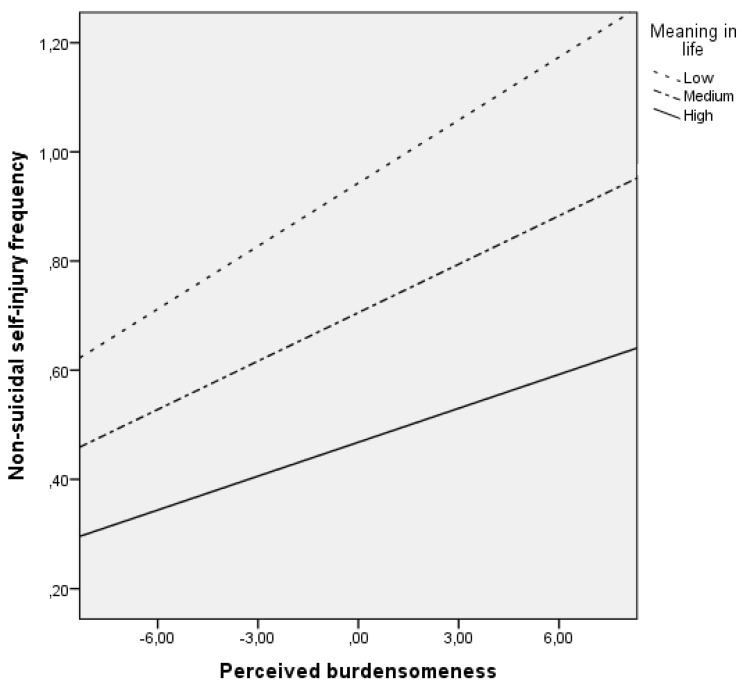
Meaning in life moderates the association between perceived burdensomeness and frequency of non-suicidal self-injury.

**Table 1 jcm-10-04867-t001:** Zero-order correlations between variables.

	*M (SD)*	2	3	4
1. Meaning in life	46.13 (9.15)	−0.61 **	−0.54 **	−0.28 **
2. Thwarted Belongingness	16.33 (8.46)		0.62 **	0.29 **
3. Perceived Burdensomeness	17.76 (6.12)			0.26 **
4. Frequency of NSSI	0.73 (1.03)			

Note. NSSI = Non-Suicidal Self-Injuries. ** *p* < 0.01.

**Table 2 jcm-10-04867-t002:** Moderation effect of MIL in the association between thwarted belongingness, perceived burdensomeness, and NSSI frequency.

Step	Variable Entered	Β	Standard Error	*t*	Total *R*^2^	Δ*R*^2^
	Variable Dependent: NSSI frequency					
1	Gender	−0.16	0.06	−2.55		
	Age	−0.03	0.02	−1.50		
2	TB	0.02	0.01	4.53 **	0.09 **	0.09 **
3	MIL	−0.21	0.01	−4.63 **	0.11 **	0.02 **
4	TB × MIL	−0.01	0.01	−3.06 **	0.12 **	0.01 **
	Variable Dependent: NSSI frequency					
1	Gender	−0.14	0.06	−2.24		
	Age	−0.04	0.02	−2.32		
2	PB	0.03	0.01	4.63 **	0.07 **	0.07 **
3	MIL	−0.02	0.01	−6.22 **	0.09 **	0.02 **
4	PB × MIL	−0.01	0.01	−1.97 **	0.10 **	0.01 **

Note. MIL = Meaning in life; TB = Thwarted Belongingness; PB = Perceived Burdensomeness; NSSI = Non-Suicidal Self-Injuries. ** *p* < 0.001.

## Data Availability

The data presented in this study are available on request from the corresponding author. The data are not publicly available due to privacy and ethical reasons.

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
