# Peer review of "Meaning in Life Buffers the Association between Perceived Burdensomeness, Thwarted Belongingness, and Frequency of Non-Suicidal Self-Injuries in Spanish Adolescents"

_jcm, 2021, doi:10.3390/jcm10214867_

Round 1
Reviewer 1 Report
The association of thwarted belongingness and perceived burdensomeness with the 4-function model is not clear to me. As there are several studies, not supporting the 4 functions.
Author Response
Dear reviewer :
Thank you very much for your kind suggestions and comments about improving the manuscript. We have addressed each point and carefully described our response after each comment. In the revised version of the manuscript, the changes are marked in blue, so that they can easily be identified by the reviewers.
Comments and Suggestions for Authors
Reviewer 1
The association of thwarted belongingness and perceived burdensomeness with the 4-function model is not clear to me. As there are several studies, not supporting the 4 functions.
In the revised version of the manuscript in the introduction we added (page 4)
When people feel that they are a burden to other family members or that they do not belong, they have deep emotional pain, which can lead to increased emotional deregulation and psychopathology. Performing NSSI would lead them to escape emotionally from this emotional pain, and NSSI would have the function of trying to bring emotional deregulation through emotional escape (Klonsky et al., 2015). Thus, the NSSI would be maintained by negative intrapersonal reinforcement. Furthermore, the relationship between thwarted belongingnes and NSSI might also be maintained by the positive intrapersonal reinforcement function. For example, people who feel unaccepted and rejected and that they do not belong to anyone or anything can perform NSSI to receive more attention from the family, from psychiatrists in the emergency service, from psychologists in the hospital admission unit, and from the nurses who take care of their wounds. All this attention and care could increase their sense of belonging. In other words, after the NSSI, their sense of belonging could be increased. Thus, theoretically, based on the four-function model of NSSI (Nock & Prinstein, 2004), the association between the feeling of thwarted belongingness and perceived burdensomeness and NSSI could be maintained by means of negative intrapersonal reinforcement and positive interpersonal reinforcement.
References:
Nock, M. K., & Prinstein, M. J. (2005). Clinical features and behavioral functions of adolescent self-mutilation. Journal of Abnormal Psychology, 114,140-146. doi: 10.1037/0021-843X.114.1.140
Reviewer 2 Report
NSSIs are an important phenomenon in the psychopathology of adolescence and require further research to identify risk factors, association with suicidal behavior, and their nosological position.The authors present the results based on a large group of adolescents, but there is no data describing this group. None data on medical history, past psychological traumas, family situation
and familial support, functioning in peer relationships and education has not been described. The authors found higher than in other studies prevalence of NSSI (27,2 % ) but did not comment
this findings. Is the prevalence related to group specificity or concommitant psychopathology or other factors? Due to limited number of colleted variables which were analyzed no hypothesis can be
verified. Due to numerous limitations mentioned by the Authors the methodology should be modified, other questionnaires should be added and the studied group should be precisley described.
Moreover, the Authors did not follow instruction for the Authors. The template was not used incorrectly.
Author Response
Dear Reviewer :
Thank you very much for your kind suggestions and comments about improving the manuscript. We have addressed each point and carefully described our response after each comment. In the revised version of the manuscript, the changes are marked in blue, so that they can easily be identified by the reviewers.
Comments and Suggestions for Authors
Reviewer 2.
1.- "NSSIs are an important phenomenon in the psychopathology of adolescence and require further research to identify risk factors, association with suicidal behavior, and their nosological position.
The authors present the results based on a large group of adolescents, but there is no data describing this group. None data on medical history, past psychological traumas, family situation and familial support, functioning in peer relationships and education has not been described.
The reviewer is right. This study is aimed at detecting risk and protective factors for NSSI in high-risk populations such as adolescents. It is a study carried out with a non-clinical sample in the natural educational context. The studies of risk factors in a non-clinical population allow us to detect the risk and protective factors of NSSI.
Because this study is designed for a non-clinical sample, we previously decided not to evaluate clinically relevant aspects, as the reviewer indicated, that are usually evaluated in studies with clinical samples.
Thus, following the indications, in the limitations section we have added (page 16):
“Fourth, we did not assess data about former or present psychopathology, suicidality, depression, psychological treatments, mental illness, pharmacotherapy, drug intake, family situation, family support, or functioning in peer relationships, and so we did not control the role of these variables in the frequency of lifetime NSSI.”
2.- "The authors found higher than in other studies prevalence of NSSI (27,2 % ) but did not comment this findings".
The reviewer is right; in the discussion section we added (page 14 )
“Regarding the frequency of lifetime NSSI, we found that 27.2% of the participants had carried out lifetime NSSI. Although there is high variability across in the frequency of NSSI across studies (Mannekote-Thippaiah, 2021), our percentages are higher than international review studies that found a prevalence of 16-18% (Muehlenkamp et al., 2012). However, our prevalence is lower than in the other study carried out in a Spanish adolescent sample (32%) (Calvete et al. 2015). This variability in the results could be due to several factors, such as the way NSSI are evaluated or the sociocultural characteristics of each sample.”
References:
Calvete, E., Orue, I., Aizpuru, L., & Brotherton, H. (2015). Prevalence and functions of non-suicidal self-injury in Spanish adolescents. Psicothema, 27(3), 223-228. https://doi.org/ 10.7334/psicothema2014.262.
Mannekote Thippaiah, S., Shankarapura Nanjappa, M., Gude, J. G., Voyiaziakis, E., Patwa, S., Birur, B., & Pandurangi, A. (2021). Non-suicidal self-injury in developing countries: A review. International journal of social psychiatry, 67(5), 472-482.https://doi.org/10.1177/0020764020943627
Muehlenkamp, J. J., Claes, L., Havertape, L., & Plener, P. L. (2012). International prevalence of adolescent non-suicidal self-injury and deliberate self-harm. Child and Adolescent Psychiatry and Mental Health, 6, 10 https://doi.org/10.1186/1753-2000-6-1
Round 2
Reviewer 2 Report
1. In reviewed manuscript all procedures were performed online. The validity of obtaining information onNSSIs by children or adolescents through self-reports is questionable, especially when additional questionnaires or tools are not used.
2. Different definitions of self-injuries can be found across current literature. The definition of NSSIs was not provided in this study. Specifically, what behaviours were considered NSSIs? The first part of the ISAS questionnaire contains 13 different methods of self-harm, including the item "other". How did the authors qualify "hair pulling, pinching, hitting yourself, scratching scabs, biting and other" as self-injuries? For example: How did the authors differentiate between pulling hair from trichotillomania, or biting themselves from biting nails? The current approach is that in order to recognize self-injury, the act of self-harm must fulfill certain specific functions. How did the authors assess the function of self-injuries? Perhaps the high percentage of self-injuries in the studied group results from the imprecise definition of the studied phenomenon?
3. Moreover, in the title of the article the Authors indicate that the study concerns Spanish adolescent, however, in a discussion the Authors conclude “Moreover, although the sample was large, it was a convenience sample, and so the results found in our study cannot be generalized to the entire population of Spanish adolescents”.
Summing up: The authors did not define precisely neither the studied phenomenon nor the group in which they studied it.
Author Response
Dear Reviewer:
Thank you very much for your kind suggestions and comments about improving the manuscript. We have addressed each point and carefully described our response after each comment. In the revised version of the manuscript, the changes are marked in blue, so that they can easily be identified by the reviewers.
Comments and Suggestions for Authors
This is a simple, reasonably designed study aimed at investigating a topic of interest for the preservation of the mental health of adolescents. There is still some minor effort to apply to improve the readability of the manuscript. Here are some suggestions.
1.-I suggest the authors change the wording of “painful events” and “painful experiences” because they can be interpreted as indicating psychological (traumatic) events or experiences. Joiner specifically refers to experience pain, physical pain, as in a self-inflicted wound. The paragraph context makes clear that the authors have this in mind, but the wording is ambiguous.
The reviewer is right; the sentence has been modified as you can see on page 3:
“the patient must develop the capability for suicide through repeated experiences of pain, such as engaging in NSSI or exposure to violence”… “the patient must acquire the capacity to do so through repeated exposure to physical pain that produces habituation to pain.”
2.-“deep emotional pain”. Again, I would say “deep emotional distress”, to avoid confusion between emotional sufferance (emotional pain) and physical pain.
Thank you for the comment; in the revised version of the manuscript, these words have been modified (page 4):
“..they experience deep emotional distress”
3.-"n= 1115, 72.8% showed 0 episodes of lifetime NSSI".
A better formulation would be: "n=1115 (72.8%) participants did not report any lifetime episode of NSSI".
Please, make the same adjustment to the description of the data by gender.
Thank you for your indications, all of which have been included in the revised manuscript.
On page 10, you can see: “n= 1115 (72.8%) participants did not report any lifetime episodes of NSSI”, “n= 587 (73.8%) participants did not report any lifetime episodes of NSSI”, “n= 528 (71.7%) participants did not report any lifetime episodes of NSSI”
"To test differences in the presence and frequency of NSSI between genders, we performed t tests and found no statistically significant gender differences in the frequency of lifetime NSSI, t(1487,051) = 1.749, p = .08".
Simplify: "There were no differences in the frequency of NSSI by gender: t(1487,05)=1.75; p=.08".
In page 11 you can see “there were no differences in the frequency of lifetime NSSI by gender, t(1487,051) = 1.749, p = .08.
4.- Did the authors use the Welch test or some other correction after violation of variance equality at Levene's test? This should be specified.
In the revised version we added (page 10).
“ We used the Welch test when there was a violation of variance equality on Levene's test”
5.-Discussion: Cut the first paragraph. The authors had already described the aims of their study.
This paragraph has been deleted
Please, comment that the explained variance (but adjusted R2 should be a better measure of it) is small, less than 10%. This comment should be not in the limitations of the study, it is a result of the study.
The reviewer is right. In the revised version, we added these sentence to the second paragraph of the discussion
However, although meaning in life was a moderator variable, the amount of variance explained by the models was small (R2 adjusted range 10-12%).
In the revised version of the manuscript, we indicate that R2 is adjusted R2 in all results. We have indicated this in the results section (page 11).
“As Table 2 shows, thwarted belongingness (t = 4.53; p< .001; adjusted R2 = .09) and perceived burdensomeness (t = 4.63; p< .001; adjusted R2 = .07)”
"[...] This is a key limitation given the lack of temporal ordering, and it is possible that the perceived burdensomeness or thwarted belongingness was a consequence of NSSI, and not the other way around.".
The authors should explain how NSSI could cause perceived burdensomeness or thwarted belongingness to give substance to their statement.
In the revised version of the manuscript, we added:
“Previous studies suggested that people with NSSI had difficulties with emotional deregulation, interpersonal relationships, or impulsivity, which lead to feelings of being a burden to their caregivers (Hoffman et al., 2003). In addition, people with NSSI showed a lack of meaningful social connections, isolation, and poor social relationships, which could make them feel thwarted belongingness (Van Orden et al., 2010).”
"The second limitation is that we only used self-report measures to evaluate the variables, and self-reports may not reflect the real constructs."
As a matter of fact, perceived burdensomeness and thwarted belongingness can be only measured at a subjective level, and the use of self-reported patient outcome measures is a good strategy for that. The same is valid for meaning in life.
The main problem is that people tend to apply the same style of responding to all questionnaires, thus a fraction of correlation depends on this rather than on a link between the examined constructs.
Thank you for your comment. The limitation has been removed from the revised version of the manuscript.
This manuscript is a resubmission of an earlier submission. The following is a list of the peer review reports and author responses from that submission.
Round 1
Reviewer 1 Report
Abstract: including gender distribution, adding to the results, what was significant.
The conclusion should be revised; this is a cross sectional study therefore it cannot be concluded what is a risk and protective factor. So far, an association has been shown with frequency, but not with other aspects of NSSI.
Introduction
Single and repetitive NSSI should be separated.
It should be further elaborated why thwarted belongingness and perceived burdensomeness are relevant in the context of NSSI and its theoretical models.
In line with this, the new aspects of the present study in addition to previous studies should be described in more detail.
The conclusion in the introduction that there are no previous studies on these associations is not correct as there are mentioned studies in the section above.
Methods:
Why was female or male an inclusion criteria?
For the assessment instruments psychometric properties have to be described. What indicates higher scores?
There was no measure assessing psychopathology and suicidality within the sample?
A major limitation is that there seems to be no control variable of depression, suicidality or psychopathology in general regarding the proposed associations.
Results:
What were the results regarding NSSI? How many participants indicated 1x, repetitive > 5x days of NSSI? A mean score is not really helpful.
Discussion:
As indicated above, the results cannot be interpreted specific for NSSI as comorbidity was not controlled for. Therefore, conclusions are considered as too exaggerated.
The adding of knowledge to the previous literature is not highlighted.
Minor issues:
- Language editing necessary
Reviewer 2 Report
This is an interesting study reporting on the association between perceived burdensomeness, thwarted belongingness, and self-injury in youth, which is a welcome addition to the literature. That said, there are several important (methodological) points the authors must address in their revision.
- The selection process lacks detail and transparency. The authors state that 1531 participants were included – but out of how many that were selected and based on what criteria? Was this at random or convenience? What was the response rate? Were there differences in characteristics between the students who participated and those who did not? Please specify.
- The authors state that participants were “college and university students” and yet their mean age was 14.85 years, ranging between 12 and 18 years. How is this possible? I don’t think adolescents aged 12 years are enrolled in Spanish universities? Again, this needs transparency.
- The authors state that “the adolescents *and/or* their parents had to give their informed consent” and “the exclusion criterion was that the adolescents *or* their parents did not agree to participate in the study – but is unclear whether both participants and their parents had to provide this – specify the “and/or” and what happened in case the minor adolescent provided consent but the parent did not. This is an important point that needs transparency. Provide how many were excluded because no consent was given. Also, specify whether this was *written* informed consent.
- Were there any safeguards in place for participants who expressed distress after participation?
- For both the PIL-A and the INQ: please give some examples of items and explicitly state the timeframe of the scales (lifetime, recent, …)
- In their analyses, the authors should definitely examine possible differential associations of the study variables between male and female participants – given the stark sex differences in NSSI (eg, https://doi.org/10.1016/S2215-0366(19)30188-9). Sex differences should be examined.
- Throughout the manuscript, it is often stated that PB or TB is a predictor of NSSI. However, the outcome measure was *lifetime* NSSI whereas the INQ refers to participants’ *recent* situation. This is a key limitation given the lack of temporal ordering (the relative timing between predictor and outcome) in the current study – it is possible that PB or TB is a *consequence* of NSSI, and not the other way around. Please be more cautious in formulations that imply a temporal relationship (eg, “perceived burdensomeness predicted the frequency of NSSI in adolescents”), and explicitly discuss this issue (lifetime vs. current assessments) in the limitations section (in addition to the generic statement that no conclusions on directionality were possible because of the cross-sectional design).
- There should be a *theoretical* discussion on how the current results fit within the IPTS of Joiner.
- The clinical implications described in the manuscript (p.7) are very limited and vague. Please elaborate on this and include more (sex-)specific recommendations. What does the literature say?